# Aflatoxicosis outbreak and its associated factors in Kiteto, Chemba and Kondoa Districts, Tanzania

**Erick Kinyenje**[1,2,3]*, **Rogath Kishimba**[2], **Mohamed Mohamed**[2], **Ambele Mwafulango**[4], **Eliudi Eliakimu**[3], **Gideon Kwesigabo**[1]

**1** School of Public Health and Social Sciences, Muhimbili University of Health and Allied Sciences, Dar es Salaam, Tanzania, **2** Tanzania Field Epidemiology and Laboratory Training Programme (TFELTP), Dar es Salaam, Tanzania, **3** Health Quality Assurance Unit, Ministry of Health, Dodoma, Tanzania, **4** National Public Health Laboratory, Dar es Salaam, Tanzania

\* kinyenje2003@yahoo.com

**Data Availability Statement:** Data related to this manuscript contains identifying or sensitive information. However, can be requested from the Permanent Secretary of Ministry of Health through

## Abstract

Tanzania had experienced hundreds of cases of aflatoxicosis in the districts of Kiteto, Chemba, and Kondoa for the three consecutive years since 2016. Cases may end up with liver cancer. Aflatoxin-induced liver cancer had resulted in the demise of roughly three persons per 100,000 in the country during the same year, 2016. We investigated to characterize the latest outbreak of 2019 and identify its risk factors. This case-control study enrolled all patients presented with acute jaundice of unknown origin and laboratory test results confirmed an acute liver injury with or without abdominal pain, distension, vomiting, or fever during the period of June to November 2019 and had epidemiological link with cases confirmed with Aflatoxin-B1-Lysine. Adjusted odds ratios (AOR) with 95% confidence intervals (CI) were used to identify independent factors associated with aflatoxicosis. We analyzed 62 cases with median age of 7 years (0.58–50 years) and 186 controls with median age of 24 years (range 0.42–55) with onset of symptoms ranging from 1st June 2019 to 16th July 2019. Case-parents had higher serum aflatoxin-B1–lysine adduct concentrations than did controls; 208.80 ng/mg (n = 45) vs. 32.2 ng/mg (n = 26); p<0.01. Storing foods at poor conditions (AOR 5.49; 95% CI 2.30–13.1), age <15 years (AOR 4.48; 95% CI 1.63–12.3), chronic illness (AOR 3.05; 95% CI 1.19–7.83) and being male (AOR 2.31; 95% CI 1.01–5.30) were significantly associated with the disease, whereas cleaning foods before milling decreased the risk of getting the disease by 88% (AOR 0.12; 95% CI 0.05–0.29). According to the results, the outbreak resulted from a globally highest-ever recorded aflatoxin-B1-lysine that originated from a common source. To prevent future outbreaks, it is crucial to store and clean food crops safely before milling. We recommend strict regulations and enforcement around aflatoxin levels in food products.

<ps@moh.go.tz> and cc to the corresponding author.

**Funding:** The authors express their gratitude for the financial assistance provided to EK and AM by the Tanzania Field Epidemiology and Laboratory Training Program (TFELTP), a program under the Ministry of Health. This assistance covered logistics and subsistence costs during data collection. The funders had no role in study design, data collection and analysis, decision to publish, or preparation of the manuscript.

**Competing interests:** The authors have declared that no competing interests exist.

## Introduction

Aflatoxicosis (AFB) is a largely ignored global health disease caused by the consumption of food sources contaminated with aflatoxins; the fungal toxins that are produced by *Aspergillus spp* [1, 2]. The disease accounts for up to 40% of all Hepatocellular carcinoma cases in Africa and 28.2% worldwide [3]. The World Health Organization (WHO) estimates that approximately 4.5 billion people in developing countries are exposed to aflatoxins, with an estimated 25% of the world's food crops being affected by aflatoxin contamination [4]. The economic impact of aflatoxin contamination is also significant, with losses estimated at several billion dollars annually due to crop losses, trade restrictions, and increased healthcare costs [5].

Hepatocellular carcinoma cost Tanzania up to $264 million annually, yet 95% of the patients died of the disease by 2014 [6]. Chronic exposure to aflatoxins could lead to but not be limited to immune suppression [7–9], malnutrition, and impaired growth in children [10, 11], a group that is more affected [12, 13].

Aflatoxins appear in fourteen different types; however, types B1, B2, G1, G2, and AFB1 are more dangerous and are found in all major food crops [14]. According to World Health Organization (WHO), exposure to AFB1 dose of 20 to 120 µg/kg body weight per day once to thrice a week is adequate to be acutely toxic and potentially lethal. East African standards set aflatoxins contamination by 10 µg/kg in food samples as a maximum tolerable limit [15]. The intoxicated case generally presents with anorexia, malaise, and low-grade fever initially and progresses to potentially fatal acute hepatitis with vomiting, abdominal pain, pulmonary oedema, convulsions, coma, and death. Studies conducted in other parts of the world have linked the disease with age [12, 13], knowledge on the disease [16], chronic consumption of alcohol, gender, and chronic illness [17].

There are five factors that contribute to aflatoxins contamination spread of aflatoxicosis; environmental conditions, agricultural practices, processing and storage practices, consumption patterns, and regulations and enforcement.

Aflatoxins thrive in warm and humid environments, and crops that are grown in such conditions are more likely to be contaminated. Additionally, poor storage conditions, such as high moisture and poor ventilation, can lead to further growth of fungi and production of aflatoxins. Furthermore, certain agricultural practices, such as the use of contaminated irrigation water, improper harvesting techniques, and inadequate pest control measures, can contribute to the spread of aflatoxicosis [3, 18, 19].

Additionally, consumption of contaminated crops is the most direct route of exposure to aflatoxins. Certain populations, such as those in low-income countries or those who rely heavily on maize and peanuts for their diet, may be at a higher risk for exposure.

Lastly, the lack of regulations and enforcement around aflatoxin levels in food products can also contribute to the spread of aflatoxicosis. Inadequate testing and monitoring of food products can lead to contaminated products entering the market and being consumed by the public [20]. Foods that have been indicated in aflatoxin contamination are peanuts, maize (corn), tree nuts, cottonseed, spices, and dried fruits [21].

Outbreaks of aflatoxicosis are rare events globally, nevertheless, Tanzania has recorded three major events in recent years 2016, 2017, and 2019 despite instituting interventions against the disease [20, 22–24]. The risk factors are yet to be studied and aflatoxin intoxications and food contamination continued to occur in several parts of the country [22, 25]. In the last two weeks of June 2019, the Tanzanian Ministry of Health received consecutive alerts on increasing cases presenting with jaundice of unknown origin, marked with elevated liver function tests. Since the cases were reported from the same three districts (Kiteto, Chemba and Kondoa) that had experienced recent aflatoxicosis outbreaks, a team was formed to investigate

the suspected aflatoxicosis outbreak and determine its associated factors. Identification of the risk factors could lead to developing and implementing informed interventions. Policymakers may use the findings from this study to establish measures that aim to minimize the risk of aflatoxin contamination and ensure the safety of food products in Tanzania.

## Materials and methods

### Ethics statement

The Ministry of Health in Tanzania approved the research plan, and the Institutional Research Ethical Committee of the Muhimbili University of Health and Allied Sciences (MUHAS) granted ethical clearance with reference number MU/PGS/SARC/Vol. IX/74. The identities of the participants were kept anonymous by using study identification numbers and laboratory serial numbers instead of their names. The data collected was securely stored in a password-protected computer and also kept in a lockable drawer as hard copies.

### Study design

We conducted an unmatched case-control study to identify potential risk factors of the outbreak. We compared patients who presented with features of aflatoxicosis to controls with no disease. Laboratory analysis of blood samples from both cases and controls was done to determine acute liver injury and quantify their exposure to aflatoxins.

### Study area

We conducted this study in three Tanzanian districts that were affected by aflatoxicosis outbreaks, namely Kiteto, Chemba, and Kondoa. These are the same districts that reported sporadic cases in years 2016, 2017 and 2019. All districts are neighbouring each other and situated in the semi-arid eco-climatic zone of the rangelands of East Africa. Having an average annual rainfall of 600-700mm and temperature of *22°C* to *27°C*, the districts attract both natives and large-scale farmers for crop production [26, 27] and as a result, the area is the giant producer of maize crop in the country [28, 29]. The wet season is warm and spans from May to November [27]. According to the National Census of 2012, the projected districts' population was 807,643 by 2019, of which about half were children below 15 years old [30].

### Study population

Study participants constituted of 62 recently identified Aflatoxicosis cases and 186 neighborhood controls identified in the study districts. We recruited cases and controls into the study using the following definitions;

### Case definition

We defined acute case of acute aflatoxicosis as any person from either Kiteto, Chemba or Kondoa District who had jaundice of unknown origin and laboratory test results confirmed an acute liver injury with or without abdominal pain, distension, vomiting, or fever during the 2019 outbreak (June to November 2019) and is epidemiologically linked to other laboratory confirmed cases. The confirmed cases were tested for aflatoxin B1-lysine albumin adducts.

### Control definition

We defined a control as any person without jaundice, abdominal pain, distension, vomiting, or fever during the period of June to November 2019 residing in a neighbourhood of the case.

## Identification of cases and recruitment of controls

We collected data through hospital document reviews, physical examination, laboratory investigation, and interviews using a structured questionnaire. We also performed a community active case search and all cases meeting the case definition were recruited. Our study physician examined the cases and then the line list was developed according to the pre-defined case definition.

We selected three controls per case from the neighbourhood of the cases. This was done by spinning a bottle at the household of the case. The first household towards the direction indicated by the bottle was picked for control number one. The bottle was spun again at the household of control number one, the fourth household towards the direction indicated by the bottle was picked for control number two. The same process was repeated to obtain control number three. The household should have not any person with symptoms or signs suggestive of aflatoxicosis, if so, the exercise was repeated. At each control's household, all family members who had slept in the house the night before were identified and listed. A lottery system was used to choose one name from the list. Papers each contained one name were folded and put in a box and shuffled well and then one was picked.

## Study instrument and data collection techniques

We used pretested questionnaires translated in the Swahili language to interview both cases and controls. For the purpose of this study; the five factors that influence aflatoxin contamination (as described above) are divided in two groups: pre-harvest factors, and post-harvest factors. Pre-harvest factors include the use of pesticides, application of farm manure, and practising crop rotation which altogether plays a key role in limiting fungal invasion and toxin production during crop growth [31, 32]. Post-harvest factors are the factors that may lead to the contamination of food crops and include food storage conditions and cleaning practices of food crops before milling [33, 34]. The questionnaire captured socio-demographic information, clinical information (for cases), knowledge on aflatoxicosis, pre-harvest factors, and post-harvest factors on aflatoxicosis. Most of the questionnaire contents were adapted from previous studies on similar contexts [22, 35].

Socio-demographic information sought from participants included age, sex, gender, occupation, and place of residence (rural or urban). Other questions were on whether participants had a chronic illness, were on chronic medications and had chronic use of alcohol (which was defined as drinking on five or more days each week) [36, 37]. Refilling the same 30-day- regimen of medication at least two times was considered a chronic use of medication [38]. The level of education attained by the head of the household and the size of the household was verified by the respective head of the household. Questions related to knowledge of aflatoxicosis were included. Knowledge questions focused on the source of aflatoxin, the risk of aflatoxin-contaminated food, and the identification of the fungi-contaminated food. Knowledge follow-up questions inquired if the respondent knew consumption of contaminated food could lead to health problems and what could be the potential consequences. Participants whose years of age was 12 and above responded on their own while parents' or guardians' knowledge was assessed on behalf of those under 12 [39, 40]. On pre-harvest factors, farmers were asked on whether they had been using fertilizers, pesticides and practising crop rotation. On post- harvest factors; we asked about the source of the consumed food prior to the onset of the symptoms (market or home-grown), the season in which the consumed food was harvested (dry or rainy), whether the food crops were cleaned before milling, the type of food storage facility used (in-house or granary) and whether the storage conditions met minimum i.e. no leakage, ventilated, clean, and dry.

## Sample size calculation

Assuming an exposure rate for food samples with aflatoxin concentration higher than the tolerable limit of 10 μg/kg was 32% among controls and 53% among cases, and a case-to-control ratio of 1:3 based upon previous studies in the same region [15, 22]; by using Kelsey et al. (1996) [41] we estimated that a sample size of 56 cases and 166 controls were required to detect a statistical difference with 80% power and 95% certainty. However, 62 cases and 186 controls met the inclusion criteria and were recruited for the study.

## Laboratory investigations

We collected about 5–10 ml blood samples from both cases and controls and tested them for liver injury makers at National Health Laboratory Quality Assurance and Training Centre (NHLQATC), Dar es Salaam by using COBAS Integra 400. Seventy-one samples from randomly sampled cases and controls were selected for the AFB1-Lysine level test at CDC Laboratory Atlanta USA. The limit of detection (LOD) for AFB1-lysine was 0.03 ng/mL. Other cases were linked by clinical presentation similarities to the laboratory-confirmed cases. On average, serum samples from cases were collected 10 days after the onset of symptoms while controls were collected after being interviewed.

## Data management and analysis

We entered, cleaned, and coded data in MS Excel 2013 and then analysed by STATA computer software version 12.0 (Stata Corp., College Station, TX, USA). We characterized the data in person, place, and time and summarized the outbreak timeline as an epidemic curve. We computed the attack rate using the formula (AR: cases/100,000 population) and the case fatality rate (CFR: deaths /cases) based on age and sex.

We provided a measure of central tendencies for continuous variables while using a Student t-test to compare aflatoxin levels among cases and controls. We performed frequency distribution and compared proportions using the Chi-square test or Fisher's exact test when the Chi-square test could not apply.

Depending on observed summary statistics and literature, age was categorized into two groups <15 and 15+years old [12], and household size was categorized as "below 5" and "above 5" based on average national household size and literature [30, 39]. Heads of households who attained at least primary education were considered to have a "formal education" and the rest had "no formal education" [42]. A participant was regarded as having good knowledge of aflatoxicosis if scored at least half of knowledge-related questions, otherwise considered poor. We used adjusted odds ratios (AOR) with 95% confidence intervals (CI) to identify independent factors associated with aflatoxicosis.

## Results

### Description of the outbreak

Fig 1 shows the epidemic curve of the outbreak in which all cases had the onset of symptoms ranging from 1st June 2019 to 16th July 2019 and 30th June 2019 was the day in which most (7) of the cases experienced the onset of symptoms. Out of 62 cases included in this study, 46 (74%) were identified through active search. The cases and controls were identical were similar at baseline.

The most frequently reported symptoms were jaundice 62 (100%), vomiting 48 (77%), abdominal swelling 40 (65%), and fever 40 (65%). The least symptom experienced by the cases

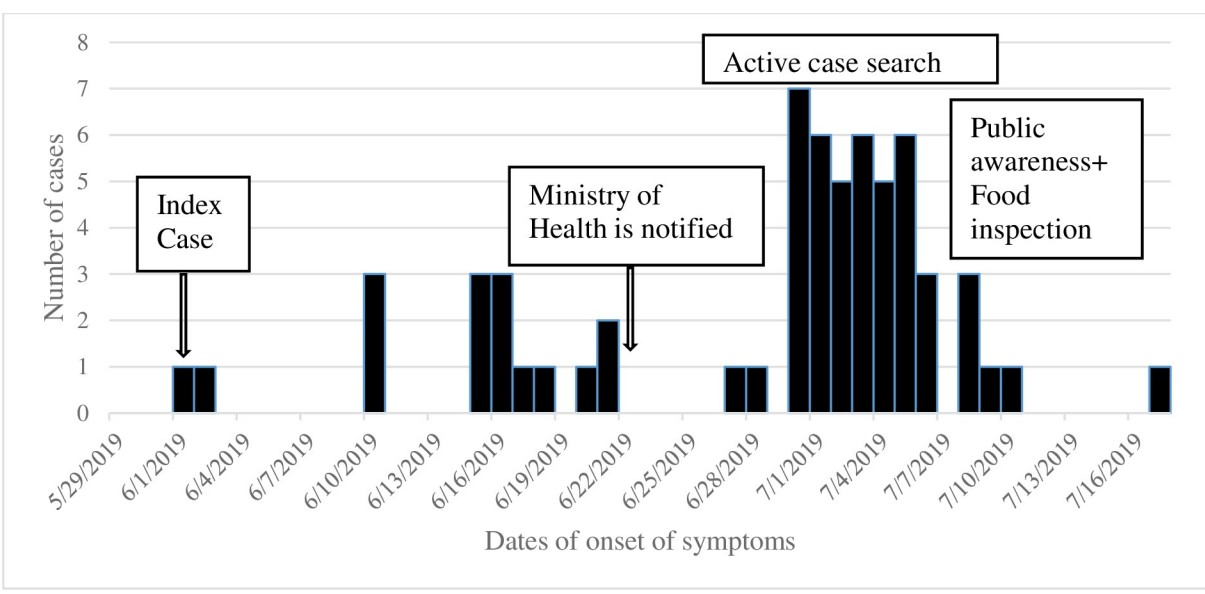

**Fig 1. Cases of aflatoxicosis (n = 62) by date of onset in Kiteto, Chemba and Kondoa Districts.**

was swelling of the scrotum 5 (8%). The median duration between the onset of symptoms and hospital admission was 5 ranging from 1 to 34 days.

The disease attack rate was highest in Kondoa District (9.9/100,000 population), followed by Chemba (8.0/100,000 population) and then Kiteto District (5.5/100,000 population). The overall case fatality rate (CFR) was 16.1%. The majority 52(83%) of cases were younger than 15 years old with relatively higher CFR (17.3%) compared to those with 15 years and above (CFR = 10%). Relatively males had a higher attack rate (9.4/100,000 population) compared to females (6.0/100,000 population), nevertheless, females had higher CFR (20.8%) compared to males (13.2%).

We recruited 248 participants for the case-control study of which all 62 cases were included plus 186 controls. Cases and controls had a median age of 7 (range = 0.58, 50) years and 24 (range = 0.42, 55) years respectively. Males predominated among cases (61.29%) while females did among controls (55.91%) $p = 0.020$. The majority of the heads of households among case-households had no formal education (69.35%) whereas the majority from control-households had formally educated (62.37%) $p <0.001$. Rural residents were predominant both among cases and controls (91.94% vs. 87.17%) and the majority of households had five or more residents (93.55% for case-households, 91.4% for control-households).

## Comparison of serum aflatoxin B1-lysine among cases and controls

We detected serum aflatoxin B1–lysine from both cases (45) and controls (26) serum specimens collected. The serum aflatoxin B1 –lysine had a median measure of 89.4ng/mL ranging from 0.395 to 719 ng/mL. Case patients tested for serum samples had six-fold mean values of aflatoxin B1–lysine adduct concentrations (208.80 ng/mg of albumin) compared to controls (32.2 ng/mg of albumin) p< 0.01.

## Factors associated with Aflatoxicosis outbreak: Findings from the case-control study

We found statistically significant associations between aflatoxicosis and being male, aged under 15 years, not having attained a formal education, being dependent for living, and having

**Table 1. Association between socio-demographic factors and aflatoxicosis in bivariate analysis.**

| Variable | Cases N = 62(%) | Control N = 186(%) | COR (95% CI) | P value |
|---|---|---|---|---|
| **Gender** | | | | |
| Male | 38(61.29) | 82(44.09) | 2.01 (1.12-3.61) | 0.020 |
| Female | 24(38.71) | 104(55.91) | Reference | |
| **Age** | | | | |
| <15 | 52(83.87) | 70(37.63) | 8.62 (4.12-18.0) | <0.001 |
| 15+ | 10(16.13) | 116(62.37) | Reference | |
| **Formal education** | | | | |
| No | 43(69.35) | 70(37.63) | 3.75 (2.02-6.94) | <0.001 |
| Yes | 19(30.65) | 116(62.37) | Reference | |
| **Place of residence** | | | | |
| Rural | 57(91.94) | 164(88.17) | 1.53 (0.55-4.23) | 0.410 |
| Urban | 5(8.06) | 22(11.83) | Reference | |
| **Occupation** | | | | |
| Farmer | 10(16.13) | 101(54.30) | 0.14 (0.07-0.30) | <0.001 |
| Businessmen | 0 | 10(5.38) | 1 | |
| Pupil/child | 52(83.87) | 75(40.32) | Reference | |
| **Size of the household** | | | | |
| Below 5 | 4(6.45) | 16(8.6) | 0.73 (0.23-2.28) | 0.59 |
| 5 and above | 58(93.55) | 170(91.4) | Reference | |
| **Chronic use of alcohol** | | | | |
| Yes | 8(12.90) | 70(37.63) | 0.25 (0.11-0.55) | 0.001 |
| No | 54(87.10) | 116(62.37) | Reference | |
| **Chronic use of medicine** | | | | |
| Yes | 15(24.19) | 26(13.98) | 1.96 (0.96-4.01) | 0.061 |
| No | 47(75.81) | 160(86.02) | Reference | |
| **Chronic illness** | | | | |
| Yes | 19(30.65) | 25(13.44) | 2.85 (1.43-5.64) | 0.003 |
| No | 43(69.35) | 161(86.56) | Reference | |

an underlying chronic illness. In contrast, chronic use of alcohol was identified as a protective factor in bivariate analysis (Table 1).

Further bivariate analysis revealed statistically significant associations between pre-harvest practices such as crop rotation, use of fertilizers, and use of pesticides with aflatoxicosis. Poor knowledge of aflatoxicosis, harvesting during the rainy season, and poor storage conditions were identified as the post-harvest risk factors for aflatoxicosis while cleaning food crops before milling was found to be a protective factor against developing the disease (Table 2).

In multivariate analysis, as presented in Table 3, living with a chronic illness, age, and gender of participants were independent predictors of aflatoxicosis among socio-demographic factors. Males were twice more likely to develop the disease compared to females [AOR = 2.26; 95% CI = (1.03, 4.96)]. Participants younger than 15 years of age were eight times more likely to develop the disease compared to those in 15 years and above [AOR = 8.49; 95% CI = (1.63, 44.3)]. Participants with chronic illness had three times higher odds of getting the disease compared to those without chronic illness [AOR = 3.05; 95% CI = (1.19, 7.83)]. At this stage of analysis; neither knowledge of aflatoxicosis, chronic use of alcohol, chronic use of medicines, nor household size had a significant association with the disease.

Participants who stored food in poor conditions were five times more likely to develop the disease compared to those stored in good condition [AOR = 5.28; 95% CI = (2.25–12.4)],

**Table 2. Knowledge, pre and post-harvest factors related to aflatoxins contamination associated with developing of aflatoxicosis in Kiteto, Chemba and Kondoa Districts.**

| Variable | Case N = 62(%) | Control N = 186(%) | COR (95% CI) | P-value |
|---|---|---|---|---|
| **Knowledge on aflatoxicosis** | | | | |
| Poor | 58(93.55) | 106(56.99) | 10.4 (3.81-31.4) | <0.001 |
| Good | 4(6.45) | 80(43.01) | Reference | |
| **Pre-harvest factors Ever farmed** | | | | |
| Yes | 13(20.97) | 109(58.60) | 0.19 (0.10-0.37) | <0.001 |
| No | 49(79.03) | 77(41.40) | Reference | |
| **Use of fertilizer** | | | | |
| Yes | 2(15.38) | 59(53.64) | 0.15 (0.03-0.74) | 0.019 |
| No | 11(84.62) | 51(46.36) | Reference | |
| **Crop rotation** | | | | |
| Yes | 2(15.38) | 19(17.27) | 1.14 (0.24-5.61) | 0.864 |
| No | 11(84.62) | 91(82.73) | Reference | |
| **Use of pesticide** | | | | |
| Yes | 5(8.06) | 53(28.49) | 0.22 (0.08-0.58) | 0.002 |
| No | 57(91.94) | 133(71.51) | Reference | |
| **Post-harvest factors** | | | | |
| **Source of food** | | | | |
| Home-grown | 59(95.16) | 177(95.16) | 1.00 (0.26-3.82) | 1.00 |
| Market | 3(4.84) | 9(4.84) | Reference | |
| **Harvest season** | | | | |
| Rainy | 38(63.33) | 69(37.91) | 2.83(1.54-5.17) | 0.001 |
| Dry | 22(36.67) | 113(62.09) | Reference | |
| **Crops cleaned** | | | | |
| Yes | 17(27.42) | 140(75.27) | 0.124 (0.06-0.24) | <0.001 |
| No | 45(72.58) | 46(24.73) | Reference | |
| **Cleaning method** | | | | |
| Dehulling | 7(41.18) | 44(42.31) | 1.02 (0.55-1.85) | 0.954 |
| Sorting | 5(29.41) | 27(25.96) | 1.22 (0.32-4.66) | 0.769 |
| Washing | 5(29.41) | 33(31.73) | Reference | |
| **Food storage facility** | | | | |
| Granary | 8(12.90) | 19(10.27) | 1.29 (0.54-3.12) | 0.565 |
| House | 54(87.10) | 166(89.73) | Reference | |
| **Storage condition** | | | | |
| Bad | 47(75.81) | 67(36.22) | 5.52 (2.87-10.6) | <0.001 |
| Good | 15(24.19) | 118(63.78) | Reference | |

while those who cleaned food crops before milling had 83% reduced risk of developing the disease compared to those who did not clean the foods [AOR = 0.17; 95% CI = (0.07, 0.38)].

## Discussion

Through this study, we aimed to describe the outbreak of acute aflatoxicosis that occurred in Districts of Kiteto, Chemba and Kondoa in 2019 and determine the association between the disease and socio-demographic, knowledge, pre-harvest and post-harvest factors.

Aflatoxins are highly toxic metabolites produced by certain strains of fungi, which can contaminate a wide range of food commodities. While the incidence of aflatoxicosis in developed countries is generally low, there have been occasional outbreaks associated with contaminated

**Table 3. Factors associated with Aflatoxicosis in multivariate analysis.**

| Variable | COR(95%CI) | P value | AOR(95%CI) | P-value |
|---|---|---|---|---|
| **Gender** | | | | |
| Male | 2.01(1.12–3.61) | 0.020 | 2.26(1.03–4.96) | 0.043* |
| Female | Reference | | | |
| **Age group** | | | | |
| <15 years | 8.62(4.12–18.04) | <0.001 | 8.49 (1.63–44.3) | 0.011* |
| +15 years | Reference | | | |
| **Formal education** | | | | |
| No | 3.75 (2.02–6.94) | <0.001 | 2.07(0.81–5.27) | 0.129 |
| Yes | Reference | | | |
| **Chronic illness** | | | | |
| Yes | 2.85 (1.43–5.64) | 0.003 | 3.05 (1.19–7.83) | 0.020 |
| No | Reference | | | |
| **Cleaned crops** | | | | |
| No | 0.12 (0.06–0.24) | <0.001 | 0.17 (0.07–0.38) | <0.001* |
| Yes | Reference | | | |
| **Storage condition** | | | | |
| Bad | 5.52 (2.87–10.6) | <0.001 | 5.28 (2.25–12.4) | <0.001* |
| Good | Reference | | | |
| **Knowledge on aflatoxicosis** | | | | |
| Poor | 10.9 (0.03–0.26) | <0.001 | 3.09 (0.79–12.1) | 0.105 |
| Good | Reference | | | |
| **Chronic use of medicine** | | | | |
| Yes | 1.96 (0.96–4.01) | 0.061 | 0.39 (0.31–1.22) | 0.107 |
| No | Reference | | | |

*Factors whose association were found significant in the final logistic regression model

food imports [43]. A review of the literature suggests that cases of aflatoxicosis in developed countries are typically associated with consumption of contaminated imported food products such as nuts, spices, and grains [43, 44]. In addition, some cases have been linked to the consumption of locally produced foods, particularly corn-based products, that have been improperly stored or processed. While there is no cure for aflatoxicosis, treatment is supportive and may involve liver transplantation in severe cases. Unlike developing countries, developed countries had employed strict regulation of food imports, monitoring of food commodities, and education of consumers about safe food handling and storage practices to prevent aflatoxicosis [44, 45].

The nature of the epidemic curve suggests the presence of an intermittent common source of outbreak whereby aflatoxin exposure could have resulted from several intoxicated food sources. We found both higher attack rate and case fatality rate among children below fifteen than those found in the previous outbreak of 2016 conducted in the same area [22]. In this regard; our findings enrich the limited evidence that aflatoxicosis affects more children below 15 [12, 13]. Future wide studies need to explore why this important group has been under attack. However, the findings from small cross-sectional studies suggest that weak immunity due to malnutrition [46] and weaning practices could be the causes [47–49].

Although the previous studies did not find significant associations between gender and high serum aflatoxin levels [17, 50]; our findings noted males were more at risk of developing the disease compared to females. This could be explained by findings obtained by Kiama T

(2016) and his colleagues that women are more cautious than males and that are less likely to eat mouldy foods compared to men [51].

The current outbreak occurred in the same season as the previous outbreaks. The recent outbreaks occurred within the same districts [22] while others occurred in the neighbouring country [12, 13]. The geographical region which has been suffering from these outbreaks is semi-arid in nature and located within 600 kilometres diameter. Furthermore, the area is characterized by the practices and climate conditions that favour the growth of fungus *Aspergillus spp*. These include but not limited to the use of maize as the common staple food [52], presence of heavy rainfall and warm weather [13, 27], previous records of aflatoxicosis outbreaks [12, 13, 17, 22, 27, 31, 53–55], and is geographically located between 40˚ N and 40˚ S [53].

In this current outbreak, the highest-ever level of serum aflatoxin globally was recorded according to the available and accessed literature [12, 17, 55–61]. The maximum serum concentration recorded in this study is about forty times that recorded in three previous years in the same districts [22] and the mean concentration of 208.8 ng/mg is even higher than the maximum value of 15.3 ng/mg recorded in that time. We suppose there has been a build-up of serum aflatoxin concentration among patients which could be due to constant exposure to contaminated foods in this area.

Chronic illness, chronic use of medications, and chronic use of alcohol have been widely documented as risk factors to aflatoxicosis [2, 17, 62, 63] through weakening the human immunity to decrease resistance against aflatoxin exposure [46, 64, 65]. It was understandable that among these three, chronic illness was an independent risk factor but not the other. Our study comprised a significant number of young participants, a population that is less likely to suffer from the risks of chronic alcohol and the effects of prolonged use of medications.

There have been contrary findings on the role of formal education in preventing aflatoxicosis. A study done in Nigeria among children found no association between serum aflatoxin levels and the disease [66] while farmers who had attained formal education were less likely to contaminate aflatoxins compared to their counterparts in DR Congo [42]. Formally educated people are said to be more conscious about the quality of the food they consume [42, 55]. We found no significant association between education level and the disease and therefore we suppose health campaigns conducted after previous attacks [13, 20, 22–24, 51, 53, 67] was the changing factor that created consciousness on the quality of food to those who lacked formal training. Our assessment on participants' knowledge was limited to reported knowledge on the disease and did not extend to knowledge on other factors such as pre and post-harvest practices. Since there was no significant association between knowledge on aflatoxicosis and the occurrence of the disease; the findings provide evidence that knowing the details of the aflatoxicosis was not enough to predict aflatoxin intoxication between the two groups, patients and controls. A compressive knowledge on multiple factors related to contamination could possibly facilitate community behaviour change, adherence to prevention practices and hence increase the adoption of the interventions and technologies available to minimize aflatoxin contamination within the community [66]. Furthermore, future studies should assess practices rather than reported knowledge to ascertain the association between the practices and the disease.

In contrast to the views of most farmers [42], our study found no significant association between practising crop rotation and the use of pesticides (among pre-harvest practices) in preventing the disease. Nevertheless, our study findings corroborate previous works that have shown the cleaning of food crops after harvest are effective in preventing aflatoxin contamination [68, 69]. Additionally, this current study is in line with previous works that conclude if foods are stored in well-ventilated facilities with no leaking roof [34], free from pests and water contamination, and in a facility that facilitates easy cleaning [16], are less likely to contaminate the aflatoxins.

## Limitations

We minimized the potential misclassification bias by testing all participants for liver injury tests (Liver function tests), and aflatoxin level tests were done to randomly selected participants from both cases and controls. We anticipated a recall bias as well and therefore respondents were asked about the information that spans not more than six months. An additional weakness of this study is the fact that we did not assess and associate socioeconomic status and the occurrence of disease. However, the available pieces of evidence have linked the two and confirmed the association between them [55, 70, 71].

## Conclusion

The findings suggest that there was a common-source outbreak of aflatoxicosis in the districts of the study caused by the global highest-ever recorded aflatoxin B1-lysine. Poor food storage practices were associated with an increased risk of aflatoxicosis, whereas, cleaning cereals before milling was a protective factor. People with underlying chronic conditions, men and children less than 15 years of age are at increased risk of getting the disease.

## Recommendations

We recommend the protection of people with underlying chronic illness and children less than 15 years of age against exposure to aflatoxins contaminations since they are at an increased risk. Local authorities should use locally available extension officers to provide community education and enforcement of regulations related to safe storage and cleaning practices of food crops before and after milling. Sectorial Ministries and stakeholders are advised to initiate a surveillance system that monitors the incidence of acute jaundice in humans to prevent widespread outbreaks of acute aflatoxicosis. Lastly, we recommend strict regulations and enforcement around aflatoxin levels in food products in the country.

## Acknowledgments

We are grateful to the study mentors, participants, the data collectors, Muhimbili University of Health and Allied Sciences and Tanzania Field Epidemiology and Laboratory Training Programme (TFELTP) for making this study possible.

## Author Contributions

**Conceptualization:** Erick Kinyenje, Rogath Kishimba.

**Data curation:** Erick Kinyenje.

**Formal analysis:** Erick Kinyenje.

**Funding acquisition:** Rogath Kishimba.

**Investigation:** Erick Kinyenje, Ambele Mwafulango.

**Methodology:** Erick Kinyenje, Rogath Kishimba.

**Supervision:** Rogath Kishimba, Mohamed Mohamed, Eliudi Eliakimu, Gideon Kwesigabo.

**Visualization:** Gideon Kwesigabo.

**Writing – original draft:** Erick Kinyenje.

**Writing – review & editing:** Erick Kinyenje, Rogath Kishimba, Mohamed Mohamed, Gideon Kwesigabo.

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
