## [Decision Letter · Decision Letter 0]

14 Mar 2023

PGPH-D-23-00013

Aflatoxicosis Outbreak and Its Associated Factors in Kiteto, Chemba and Kondoa Districts, Tanzania

Dear Dr. Kinyenje

Thank you for submitting your manuscript to PLOS Global Public Health. After careful consideration, we feel that it has merit but does not fully meet PLOS Global Public Health’s publication criteria as it currently stands. Therefore, we invite you to submit a revised version of the manuscript that addresses the points raised during the review process.

Please respond to all the reviewers comments and resubmit your manuscript for consideration for publication. 

Please ensure that your decision is justified on PLOS Global Public Health’s publication criteria and not, for example, on novelty or perceived impact.

We look forward to receiving your revised manuscript.

Kind regards,

Reuben Kiggundu

Academic Editor

Journal Requirements:

Additional Editor Comments (if provided):

Reviewers' comments:

Reviewer's Responses to Questions

**Comments to the Author**

1. Does this manuscript meet PLOS Global Public Health’s publication criteria? Is the manuscript technically sound, and do the data support the conclusions? The manuscript must describe methodologically and ethically rigorous research with conclusions that are appropriately drawn based on the data presented.

Reviewer #1: Partly

Reviewer #2: Yes

2. Has the statistical analysis been performed appropriately and rigorously?

Reviewer #1: Yes

Reviewer #2: Yes

3. Have the authors made all data underlying the findings in their manuscript fully available (please refer to the Data Availability Statement at the start of the manuscript PDF file)?

Reviewer #1: No

Reviewer #2: No

4. Is the manuscript presented in an intelligible fashion and written in standard English?

Reviewer #1: Yes

Reviewer #2: Yes

5. Review Comments to the Author

Reviewer #1: I think the authors made a great choice in terms of their starting point. I'm not convinced by the literature, though.

As a first step, the author should inform readers about the background of the disease, what factors contribute to its spread, and the worldwide perspective on the issue. Therefore, they should focus solely on saving the lives of Africans in imminent danger from this should be pointed. then references to the relevant literature from the region should be made to narrow the scope of this investigation.

Writers owe it to their audiences to inform them about the background of their neighborhood of choice, e.g., 4 specific districts.

Author should make connection from developed world literature to how this outbreak may have been avoided through proper storage and cleaning of food crops prior to processing, and how public health policies could have assisted in this endeavor.

The author should mention these factors in the implications section because we cannot alter demography or lifestyles but can suggest policies that may lead to systemic prevention in the future.

The discussions and conclusions are oversimplified and need to be rewritten in light of the literature and building sociological and scientific argument. Readers interested in public health from all around the world (developing, non-developing, disease-free, infected) should get something from this article.

Figure 1 is of low quality.

Reviewer #2: Thank you for an informative manuscript dealing with a rare but important condition. However, the authors need to address the following.

Background

a) Include food crops that have been implicated in aflatoxicosis.

Results

a) What proportion of cases was identified through active search?

b) Were cases and controls similar at baseline?

6. PLOS authors have the option to publish the peer review history of their article (what does this mean?). If published, this will include your full peer review and any attached files.

**Do you want your identity to be public for this peer review?** For information about this choice, including consent withdrawal, please see our Privacy Policy.

Reviewer #1: No

Reviewer #2: **Yes: **Addmore Chadambuka

---

## [Editor Report · Decision Letter 1]

18 Jul 2023

Aflatoxicosis Outbreak and Its Associated Factors in Kiteto, Chemba and Kondoa Districts, Tanzania

PGPH-D-23-00013R1

Dear Dr. Kinyenje

We are pleased to inform you that your manuscript 'Aflatoxicosis Outbreak and Its Associated Factors in Kiteto, Chemba and Kondoa Districts, Tanzania' has been provisionally accepted for publication in PLOS Global Public Health.

Best regards,

Reuben Kiggundu

Academic Editor
